# Respiratory Variations of Central Venous Pressure as Indices of Pleural Pressure Swings: A Narrative Review

**DOI:** 10.3390/diagnostics13061022

**Published:** 2023-03-07

**Authors:** Michele Umbrello, Sergio Cereghini, Stefano Muttini

**Affiliations:** SC Terapia Intensiva Neurochirurgica, ASST Santi Paolo e Carlo Polo Universitario, Ospedale San Carlo Borromeo, Via Pio II, 3, 20151 Milano, Italy

**Keywords:** esophageal pressure, central venous pressure, inspiratory effort, lung mechanics, pleural pressure, critically ill patients

## Abstract

The measurement of pleural (or intrathoracic) pressure is a key element for a proper setting of mechanical ventilator assistance as both under- and over-assistance may cause detrimental effects on both the lungs and the diaphragm. Esophageal pressure (Pes) is the gold standard tool for such measurements; however, it is invasive and seldom used in daily practice, and easier, bedside-available tools that allow for rapid and continuous monitoring are greatly needed. The tidal swing of central venous pressure (CVP) has long been proposed as a surrogate for pleural pressure (Ppl); however, despite the wide availability of central venous catheters, this variable is very often overlooked in critically ill patients. In the present narrative review, the physiological basis for the use of CVP waveforms to estimate Ppl is presented; the findings of previous and recent papers that addressed this topic are systematically reviewed, and the studies are divided into those reporting positive findings (i.e., CVP was found to be a reliable estimate of Pes or Ppl) and those reporting negative findings. Both the strength and pitfalls of this approach are highlighted, and the current knowledge gaps and direction for future research are delineated.

## 1. Introduction

The correct setting of ventilator assistance is a daily challenge in the care of critically ill patients [1,2,3,4]. Both controlled and assisted ventilation can cause injury by different interacting pathways; lung damage is mainly mediated by the mechanical stress and strain generated in the processes referred to as ventilator-induced lung injury (VILI) [5] and patient self-inflicted lung injury (P-SILI) [6], while diaphragm harm depends on myotrauma that is due to under- or over-support, which respectively lead to load-induced muscular injury [7] and disuse atrophy [8]. Especially during the weaning phase, critically ill patients are at risk of vigorous inspiratory efforts because of augmented respiratory drive, altered lung mechanics or inadequate sedation [9], and this, in turn, is a strong predictor of weaning failure [10].

Transpulmonary pressure (P_L_) is the difference between airway pressure (Paw) and pleural (Ppl) pressure and represents the total lung stress acting on the lung parenchyma; it is generated by both the ventilator and the patient [11]. During assisted ventilation, strong inspiratory efforts can increase P_L_, acting on the negative pleural pressure swings and potentially leading to a non-protective ventilation that cannot be assessed with the conventional ventilator waveforms. In addition, these inspiratory efforts can amplify the transmural pulmonary vascular pressure swings, worsening vascular leakage and causing weaning-induced pulmonary edema [12]. Therefore, the setting of ventilatory assistance must aim to minimize P_L_ in order to avoid P-SILI and diaphragm injury while maintaining a proper level of patient effort to avoid atrophy and promote spontaneous breathing [13].

Therefore, it seems intuitive that the close monitoring of partitioned lung mechanics and patient respiratory drive and effort is a key point to reduce the iatrogenic harm of mechanical ventilation as much as possible. Unfortunately, the gold standard tools for such measurements are invasive and seldom used in daily practice, and easy, bedside-available tools that allow for rapid and continuous monitoring are greatly needed. The tidal swing of central venous pressure (ΔCVP) has long been proposed as a reasonable surrogate for the pleural pressure; however, despite the widespread availability of central venous catheters in the critical care setting, this variable is very often overlooked. In the current paper, we describe the physiological base and systematically review the evidence regarding the use of central venous pressure waveforms to estimate pleural pressure, highlighting both the strength and pitfalls of this approach and outlining the current knowledge gaps.

## 2. Monitoring Patient Respiratory Effort

An adequate evaluation of spontaneous respiratory activity from physical examination or ventilator waveforms is often difficult. Therefore, several invasive and noninvasive methods have been developed, each with its strengths and weaknesses.

The esophageal pressure (Pes) measurement is considered the gold standard estimate of Ppl and has been extensively studied both to obtain partitioned respiratory mechanics as well as to assess patient respiratory activity and work of breathing [14]. The simplest parameter to estimate spontaneous effort is the amplitude of its negative inspiratory deflection, and a target ΔPes of approximately 3–8 cmH_2_O has been suggested during assisted ventilation [13]. The analysis of the Pes waveform also allows for the calculation of other more complex measurements, such as inspiratory muscle pressure (Pmus) and the pressure-time-product (PTP_es_), discussed in detail elsewhere [15]. However, the use of Pes remains infrequent in the clinical setting, likely because it requires some technical and physiological expertise when positioning the catheter and interpreting the pressure swings [16].

Another interesting tool to evaluate respiratory drive and effort is the electrical activity of the diaphragm (EAdi). The electromyographic signal represents the electrical activation of the crural diaphragm, which is proportional to the stimulus of the phrenic nerve and, therefore, closely reflects respiratory motor output. Moreover, maximum EAdi during tidal breathing (EAdi_peak_) can be used to measure inspiratory effort since it strongly correlates with Pes [17]. A wide inter-individual variability in EAdi normal values (5–30 µV) is the prevalent limitation of this technique [18], which makes any specific EAdi threshold during assisted ventilation difficult to target, although still being a very useful tool to evaluate patient–ventilator interaction and the trend of spontaneous ventilatory activity [19].

In regards to the noninvasive techniques, expiratory and inspiratory occlusion maneuvers can also be used to estimate the inspiratory drive and effort. In particular, the airway occlusion pressure (P_0.1_) is the Paw reduction during the first 100 ms of an expiratory occlusion and provides a measure of the patient’s respiratory drive, being independent of pulmonary mechanics and diaphragm function [20]. A P_0.1_ target value during assisted mechanical ventilation between 1.5 and 3.5 cmH_2_O has been proposed to titrate support [21]. The Pmusc index (PMI) [22] is the estimate of the pressure developed by inspiratory muscles during an inspiratory effort; it is calculated as the difference between the elastic recoil pressure (Pel,rsi) at the end of an inspiratory occlusion maneuver and the sum of positive end-expiratory pressure (PEEP) and pressure support. While being reliable tools to evaluate a patient’s inspiratory effort and avoid injurious breathing, these indices still have the intrinsic limitations of a static and discrete measurement.

Diaphragm ultrasound similarly represents a useful noninvasive exam that permits a good estimate of the inspiratory effort. In particular, the diaphragm thickening fraction (TFdi) correlates well with EAdi [23] and esophageal and diaphragmatic pressure-time products [24]. However, even this technique has limitations that include operator dependency, non-continuous acquisition, influence of thoracoabdominal breathing pattern and need for adequate equipment. For these reasons, the diaphragm ultrasound is best used for intermittent patient assessment, and it is not widespread for spontaneous inspiratory effort evaluation in the clinical setting.

## 3. Central Venous Pressure Swings: Physiological Background

During ventilation, the changes in both lung and pleural pressures are naturally transmitted to the other structures enclosed in the mediastinum, including the cardiovascular system. Since the heart can be considered a pressure chamber within a pressure chamber, it is known how intrathoracic (or pleural) pressure affects circulatory pressure gradients and hemodynamics [25]. Furthermore, the superior vena cava is a highly compliant intrathoracic vein, which explains why the intrathoracic pressure changes that occur during both mechanical and spontaneous ventilation have a significant impact on CVP values and contribute to their respiratory swings. Previous physiological investigations well described the phasic changes associated with ventilation in heart rate [26], autonomic tone [27], pulmonary vascular resistance [28] and venous return [29].

In particular, positive pressure mechanical inspiration increases intrathoracic pressure, thus causing a positive ΔCVP that reduces venous return. Conversely, spontaneous inspiration decreases intrathoracic pressure, thus causing a negative ΔCVP that increases venous return [30]. Indeed, this increase in venous return is limited; when the pressure drops below atmospheric pressure, the great veins collapse and develop a flow limitation. Therefore, based on these physiological assumptions, CVP appears to have respiratory oscillations that reflect intrathoracic (i.e., pleural) pressure changes. The extent of the respiratory swing in CVP was shown to be of a similar extent to that of Pes, which is also justified by the close anatomical position of the two measurement systems (as depicted in Figure 1 and Figure 2, upper panel).

In summary, CVP may represent a valid estimate of Ppl, similar to Pes, with the undeniable advantage of an almost ubiquitous presence of a central venous catheter in the critical care setting. This can provide an easy and rapid tool to evaluate partitioned respiratory mechanics and patient contribution during assisted breathing and guide titration of ventilatory support. The lower panel of Figure 2 shows the simultaneous recordings of flow, airway, esophageal and central venous pressures in a critically ill patient undergoing assisted mechanical ventilation. The use of ΔCVP as a surrogate of ΔPpl has been examined for decades, although it never became popular in the care of critically ill patients. In the following paragraphs, we summarize the findings of previous and recent papers that addressed this topic; the studies are divided into those reporting positive findings (i.e., CVP was found to be a reliable estimate of Pes or Ppl) and those reporting negative findings. Table 1 summarizes the factors influencing the relationship between CVP and pleural pressure; Table 2 reports the setting, methods and main findings of the studies included in the present review, and Table 3 summarizes the main biases and the limits of agreement of the different studies.

## 4. Clinical Studies

### 4.1. Studies in Which CVP Was Found to Be a Reliable Estimate of Ppl

Despite the robust physiological basis for the use of CVP to estimate Ppl and the widespread availability of central venous catheters in critically ill patients, only a few clinical studies investigated the role of CVP respiratory swings to monitor Ppl variations. These analyses were conducted under three different main ventilation settings: controlled mechanical ventilation, spontaneous breathing and assisted mechanical ventilation.

During controlled mechanical ventilation, the estimation of ΔPpl from ΔCVP allows an easy and rapid monitoring of P_L_ that is a key component of lung protective ventilation [16]. Walling and Savege [31] calculated P_L_ from both ΔPes and simultaneous changes measured from ΔCVP in supine patients during controlled positive-pressure ventilation. They enrolled nine subjects who were studied on 12 different occasions and found that the respiratory swings of P_L_ measured from ΔPes were on average 30% higher than those measured from ΔCVP. Notably, previous studies reported a similar higher ΔPes (26.4%) as compared with that from the direct measurement of ΔPpl [45], suggesting a comparable value of ΔPpl and ΔCVP. Moreover, despite the different absolute value, the authors also reported a linear correlation between CVP- and Pes-derived Pl (R = 0.74, *p* < 0.001). The main limitations of that study were that CVP was measured at different sites and with different catheters, and fluid-filled systems (central venous catheter) were compared with air-filled systems (esophageal catheter) of different lengths and diameters, thus with different frequency response characteristics that could distort the signals. Moreover, the authors did not use any occlusion test [46] to find the proper position of the esophageal catheter and only checked it radiographically.

A more recent study by Kyogoku et al. [32] aimed to develop a correction method for estimating respiratory ΔPpl by using ΔCVP. The analysis was conducted on seven children with acute respiratory failure, paralyzed and mechanically ventilated; the tidal ΔPes, ΔCVP and ΔPpl that were calculated using a corrected ΔCVP (κ × ΔCVP, where κ was the ΔPaw/ΔCVP ratio when compressing the thorax in the occlusion test) were compared. This corrected ΔPpl (cΔCVP-derived ΔPpl) correlated with ΔPes better than did ΔCVP (R^2^ = 0.48, *p* = 0.083 vs. R^2^ = 0.14, *p* = 0.407), thus improving the accuracy of ΔPpl estimation by using ΔCVP. The findings were limited by the small sample size, the exclusion of severe ARDS cases and the inclusion of only young patients who mostly underwent cardiac surgery with important effects in terms of chest wall elastance. Consequently, the results may not be generalizable, and the correction method may be prone to error in adults.

Okuda et al. [33] used the same method in a recent preliminary study that examined eight children who were mechanically ventilated under 10, 5 and 0 cmH_2_O of pressure support. Similar to the previous study, in this analysis, the ΔCVP-derived ΔPpl was calculated using a corrected ΔCVP (κ x ΔCVP, where κ was the ΔPaw/ΔCVP ratio during a spontaneous inspiration against an occluded airway). The difference of the cΔCVP-derived ΔPpl to ΔPes was smaller than that of ΔCVP to ΔPes at all support levels (−0.1 ± 1.5 vs. 3.1 ± 3.5 cmH_2_O in PS 10, −0.7 ± 3.3 vs. 4.5 ± 3.9 cmH_2_O in PS 5 and −1.0 ± 3.4 vs. 4.7 ± 4.4 cmH_2_O in PS 0). Moreover, the repeated measures correlation between cΔCVP-derived ΔPpl and ΔPes indicated that the former had a better correlation with ΔPes (r = 0.84, *p* < 0.0001). Despite the numerous limitations already mentioned for the previously analyzed article, it seems that this ΔCPV correction method allows a reliable evaluation of ΔPpl without using an esophageal balloon, at least in children under assisted mechanical ventilation.

Another recent investigation by Verscheure et al. [34] compared respiratory-induced changes in pulmonary artery occlusion, central venous and esophageal pressures during pressure-regulated volume control (PRVC) and pressure support ventilation (PSV). A total of 30 patients who underwent elective cardiac surgery and had pulmonary artery and fluid-filled esophageal catheters in place were enrolled. During PRVC, the maximum inspiratory increase in pulmonary artery occlusion pressure (Paop) and CVP were compared with the maximum increase in Pes; the bias for ΔCVP was, on average, 0.3 mmHg with limits of agreement 2.8 to −2.1 mmHg. During PSV, the maximum negative deflections in Paop and CVP were compared with the maximum negative deflections in Pes. The bias for ΔCVP was −2.2 mmHg (limits of agreement 1.4 to −5.8 mmHg), likely because transmural CVP increased in many subjects. These data confirmed that a fluid-filled esophageal catheter reliably tracks both positive and negative changes in Ppl as indicated by ΔCVP.

The evaluation of CVP swings and their correlation with Ppl variations during tidal ventilation were also investigated in spontaneously breathing patients. Flemale et al. [35] compared ΔCVP, ΔPes and mouth occlusion pressure respiratory changes (ΔPm) as an estimate of ΔPpl in 10 healthy adults in different body positions during inspiratory efforts against occluded airways. In that study, ΔPm during the occlusion test was considered representative of ΔPpl according to previous findings [46,47]. Pm, Pes and CVP were measured using similar catheters, and each system was filled with water to minimize errors and biases. The ΔCVP/ΔPm, ΔPes/ΔPm and ΔPes/ΔCVP individual values and group average were close to unity in all positions. The authors concluded that the central venous catheter and the water-filled esophageal catheter, when validated with the occlusion test, can provide, in most instances, accurate measurements of ΔPpl.

Another interesting but slightly different analysis was recently conducted by Aguilera et al. [36]. Cough pressure as an expression of expiratory muscle strength was recorded in nine patients using Pes as the gold standard and compared with gastric (Pga), CVP, bladder (Pbl) and rectal pressures (Prec). The average maximum pressures at those different sites were similar, and an excellent agreement was found between alternative sites and Pes (only Pbl was slightly higher than Pes). Besides further validating the use of less invasive catheters to measure esophageal pressure, that paper demonstrates that intrathoracic pressure changes are well transmitted to the vascular compartment and, in particular, to the SVC.

In regards to patients undergoing assisted mechanical ventilation, the ability to estimate ΔPpl using ΔCVP could help to monitor the spontaneous inspiratory effort and, consequently, titrate pressure support [48]. Chieveley-Williams et al., investigated whether ΔCVP and ΔPbl may reflect ΔPes and ΔPga and whether their changes were comparable when inspiratory support was modified in 10 patients under pressure support ventilation [37]. Pressure support was progressively reduced by 5 cmH_2_O steps until zero or the minimum tolerated. At the lowest level of pressure support, the ΔPes/ΔCVP ratio varied between 0.8 and 2.1, and the ΔPga/ΔPbl ratio varied between 0.6 and 1.3. Moreover, albeit absolute values of ΔCVP did not reflect ΔPes during assisted mechanical ventilation, after reduction of the ventilator assistance, the variation in ΔCVP was related with the variation of ΔPes, suggesting that ΔCVP can be a rapid guide for ventilatory support titration.

Biselli and Nobrega [38] compared ΔCVP and ΔPes for the measurement of work of breathing (WOB), effort and lung mechanics in 10 patients under assisted ventilation. ΔCVP was highly correlated with ΔPaw during spontaneous inspiration with an occluded airway (Muller maneuver), and this correlation was comparable to that between ΔPes and ΔPaw. Furthermore, CVP showed a good performance for measuring WOB (r^2^ = 0.89) and intrathoracic pressure swings (r^2^ = 0.75) when compared to Pes. Similar conclusions were reported in a recent study by Colombo et al. [39]. Assuming that ΔCVP may reflect ΔPpl, ΔPes and, therefore, the strength of inspirations, the authors aimed to determine the diagnostic accuracy of ΔCVP for strong inspiratory efforts (arbitrarily defined as ΔPes >8 mmHg). ΔCVP and ΔPes were measured in 48 critically ill patients undergoing spontaneous breathing with zero (ZEEP) or 10 cmH_2_O CPAP. ΔCVP identified strong inspiratory efforts with an area under the curve of >0.9 both at ZEEP and during CPAP, and a good agreement and correlation were reported between ΔCVP and ΔPes. Notably, however, ΔCVP and ΔPes frequently diverged by more than 20% in many patients. These findings suggested that ΔCVP, even if sometimes differing from the ΔPes exact value, could help to identify strong inspiratory efforts and eventually modify ventilatory support.

Similar findings were reported in a very recent study by Lassola et al. [40], which examined the comparative performance of diaphragm ultrasound and CVP tidal swings as a measure of inspiratory effort in 14 critically ill patients with COVID-19 undergoing a three-level pressure support trial (10, 5 and 0 cmH_2_O). By reducing support from 10 to 0 cmH_2_O, both ΔPes and ΔCVP increased in a similar way (5 [3; 8] vs. 8 [14; 13] vs. 12 [6; 16] and 4 [3; 7] vs. 8 [5; 9] vs. 10 [7; 11] cmH_2_O, respectively). Furthermore, ΔCVP was significantly associated with ΔPes with a coefficient of determination (R^2^ = 0.810, *p* < 0.001) higher than that between diaphragm thickening and ΔPes (R^2^ = 0.399, *p* < 0.001). Notably, the association between ΔCVP and ΔPes was similar in both patients with high or low CVP, considering 14 cmH_2_O (median value of end-expiratory CVP) as a cutoff (R^2^ = 0.862, *p* < 0.001 and R^2^ = 0.817, *p* < 0.001, respectively). Eventually, ΔCVP could discriminate high inspiratory efforts (arbitrarily defined as ΔPes > 8 cmH_2_O) even better than diaphragm TR, again suggesting how the tidal swing of CVP might be a rapid bedside tool to monitor patient inspiratory effort during assisted ventilation.

### 4.2. Studies Which Concluded That CVP Cannot Reliably Estimate Ppl

Indeed, the use of ΔCVP as a surrogate of ΔPpl is not universally accepted. Notably, the literature also reports conflicting data on CVP respiratory swings showing, in some instances, a poor correlation with ΔPpl or ΔPes. Ostrander et al. [41] compared tidal CVP changes and directly measured Ppl respiratory variations in 10 supine dogs. During normal breathing, ΔPpl was transmitted to the vena cava with attenuation, which was even amplified with increasing mean CVP. In fact, ΔCVP was approximately 55% of ΔPpl at low mean CVP and decreased to 20% of ΔPpl at high CVP values in contrast to previous observations [31]. Another finding was a temporal delay in ΔCVP compared to ΔPpl, which increased as the mean CVP became higher. This was suggested to have several potential causes: a delay in transmission of pressures from the intrapleural space to the vena cava, unequal delays in the transducing of electrical signals or nonhomogeneous pressure distribution in the lungs. Finally, the authors described the addition of a cardiac component that should be removed with electronic filters to reduce measurement errors. Thus, these findings suggested caution when attempting to evaluate absolute values of ΔPpl directly from ΔCVP without appropriate waveform corrections, though these two parameters were clearly correlated.

Hedstrand et al. [42] described some important distortions when comparing respiratory ΔCVP with ΔPes in 13 healthy subjects studied in supine, semi-recumbent and sitting positions. First, a strong mean phase divergence between ΔCVP and ΔPes of approximately 180° with large intra- and inter-individual variations and no effect of body position was found. Only the application of an external airway resistance reduced this lag. In addition, the mean ΔCVP/ΔPes ratio varied significantly among the individuals and with body position, ranging from 0.28 (0.12–0.44) in the supine, 0.42 (0.25–0.56) in the semi-recumbent and 0.68 (0.34–1.40) in the seated positions. Even in this case, the application of an external airway resistance reduced the inter-individual and positional variations. The authors cautioned against a simple replacement of ΔPes with ΔCVP when estimating ΔPpl, despite the fact that the presence of increased airway resistance improved the accuracy of the findings.

The correlation between ΔCVP and ΔPes was investigated in a study aimed at calculating lung compliance with both variables during mechanical ventilation in 12 patients with acute respiratory failure [43]. ΔPes tended to be higher than ΔCVP, and the measurements were only weakly correlated (r = 0.47). However, the values of ΔP_L_ and lung compliance calculated with both methods correlated well (r = 0.94 and r = 0.91, respectively), probably because of the high value of ΔPaw, not comparable to the smaller ΔPes and ΔCVP values. Again, these findings confirm that ΔCVP does not exactly replicate absolute values of ΔPes.

Likewise, a more recent study by Bellemare et al. [44] found a low correlation between ΔCVP and ΔPes during spontaneous inspiratory efforts in 24 intubated patients; the bias between the two measurements was 2.9 cmH_2_O with R^2^ = 0.43. The authors argued that the reason might lie in the different volumes that fill the right and left hearts. In particular, the left heart venous reservoir is contained in the chest and is affected by the same pressure changes that affect the left atrium, while the right heart venous reservoir is mainly outside the chest, and the fall in Ppl increases the pressure gradient from the systemic venous reservoir to the right atrium.

### 4.3. Open Issues and Future Developments

As outlined in the previous paragraphs, many relevant unanswered questions remain for the use of ΔCVP as a rapid and bedside surrogate of ΔPpl. First, two different systems are compared: one filled with fluid (central venous catheter) and one filled with air (esophageal balloon). Only one of the quoted studies [35] analyzed ΔCVP and ΔPes with the same fluid-filled system and showed that the central venous catheter as compared to a fluid-filled esophageal catheter provides a valid assessment of ΔPpl in healthy adults during spontaneous breathing. Nevertheless, in the more recent studies where ΔCVP and ΔPes were measured with different systems, a correlation was consistently found between these two parameters. This probably depends on the fact that, over time, transmission and transduction systems have become more accurate, minimizing errors and biases. While it is well-accepted that fluid-filled systems have a better frequency response to pressure variations, no large differences have been demonstrated when comparing fluid-filled and air-filled systems for ΔPes monitoring [49]. From a practical standpoint, we suggest analyzing the CVP trace using the “wedge pressure procedure” available in almost any standard intensive care multiparametric monitor. In fact, this procedure displays an invasive pressure trace together with the respiration waveform as a reference and allows the clinician to freeze the trace and move the cursor while displaying the actual pressure value, allowing for an accurate selection of the inspiratory and expiratory values of CVP relative to the respiratory cycle.

Indeed, in many studies, the agreement between ΔCVP and ΔPes was not accurate enough to suggest using these parameters interchangeably. The relative position of the esophageal balloon [50] and the central venous catheter in the thorax may account for some of the lack of concordance between these two different techniques [51]. Another element that can influence the association between ΔCVP and ΔPes is the underlying pathology. In fact, different pathophysiological alterations (such as inflammation, edema and fibrosis) or dynamic respiratory mechanisms (such as air-trapping, alveolar gas compression and decompression) could lead to a different transmission of the intrathoracic pressure into the vascular and gastrointestinal systems, thus altering the correlation between ΔCVP and ΔPes [48,50]. Further studies are needed to understand if and when the estimate of ΔPpl is better reflected by ΔPes or ΔCVP. However, even if their absolute values might not be equal, it is generally possible to compare the ΔCVP trend with ΔPes to monitor inspiratory efforts, especially during the reduction in pressure support in assisted mechanical ventilation [37,39,40].

Another issue depends on the presence of cardiac artifacts or the shape of the peaks and troughs of the CVP waveform that could potentially impact the ΔCVP evaluation. Indeed, these interferences can easily be reduced with signal filtration using low-pass or novel time-variant filtering techniques [52], similar to what happens with Pes, and by using appropriate CVP lecture methods [53]. On the other side, any alteration of the cardiac rhythm or cardiac output to vascular system coupling could possibly influence the CVP waveform, making any derived measurements unreliable.

A key aspect that is not completely understood is patient volume status, which from a theoretical point of view could heavily influence the measurement of tidal ΔCVP. Indeed, the vena cava has intrinsic elastic characteristics that could be affected and modified by fluid overload or a positive fluid balance. In particular, the filling state of the vein could influence its compliance with the result of a poor transmission of the pleural pressure to the vascular structure. Ostrander et al. [41] probably observed this effect in their physiological study. They showed how respiratory ΔPpl was transmitted to the vena cava with an attenuation, which was even amplified with a higher mean CVP. This was in contrast to Walling and Savage’s findings [31] of a greater overlap between tidal ΔCVP and ΔPes with a higher mean CVP (>10 cmH_2_O). More recent studies [39,40] did not indicate any statistically significant effect of a high mean CVP value or hypervolemia on the correlation between ΔCVP and ΔPes, but enrolment issues might have masked this effect. Furthermore, a high mean CVP was not associated with a reduction in the diagnostic power of ΔCVP to detect high inspiratory efforts during assisted mechanical ventilation. Expanding on this point, since CVP is dependent upon the interaction of heart function and return function [54], not only the volume status but also the cardiac function (and especially the right heart function) might influence the respiratory fluctuation in central venous pressure. While no studies are available in the literature that specifically address this issue, it is conceivable that an elevated transpulmonary pressure in the presence of a failing right ventricle leads to an augmented right ventricular afterload [55] that could, in theory, increase CVP and potentially influence its values irrespective of the inspiratory effort.

Given these uncertainties, it is clear that further insights and research are needed to understand the impact of the filling status and right heart function or the mean CVP value on tidal ΔCVP reliability in respiratory ΔPpl evaluation, especially at the extremes of filling volumes and cardiac function.

## 5. Conclusions

The tidal swing of CVP seems to be an easily available and reasonable surrogate for ΔPes during both mechanical ventilation and spontaneous breathing. Even if their absolute values might not be completely overlapping, CVP and Pes fluctuate in a similar way during ventilation, and it could be useful to compare their trend to monitor respiratory Ppl variations. Especially during assisted ventilation, ΔCVP can be used as a rapid bedside tool to evaluate the patient’s contribution to the total work of breathing. This could help to better titrate the ventilatory support and avoid both dangerous inspiratory efforts and disuse muscular atrophy. The CVP measurement systems, the waveform artifacts filtration, the patient filling status and, possibly, right heart function influence on ΔCVP and the lack of outcome studies represent still open questions that require further investigations.

## Figures and Tables

**Figure 1 diagnostics-13-01022-f001:**
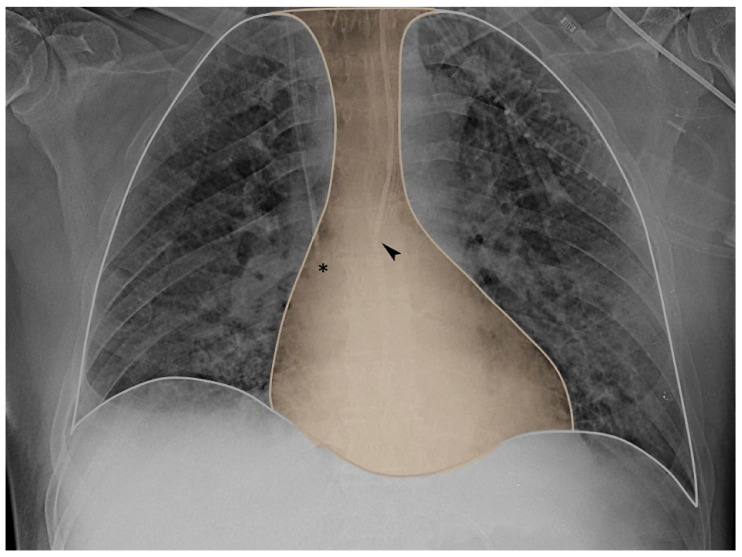
Chest X-ray mediastinum view. The tip of the central venous catheter (asterisk) and the tip of the esophageal balloon (arrow) are both closely located within the mediastinum (yellow area).

**Figure 2 diagnostics-13-01022-f002:**
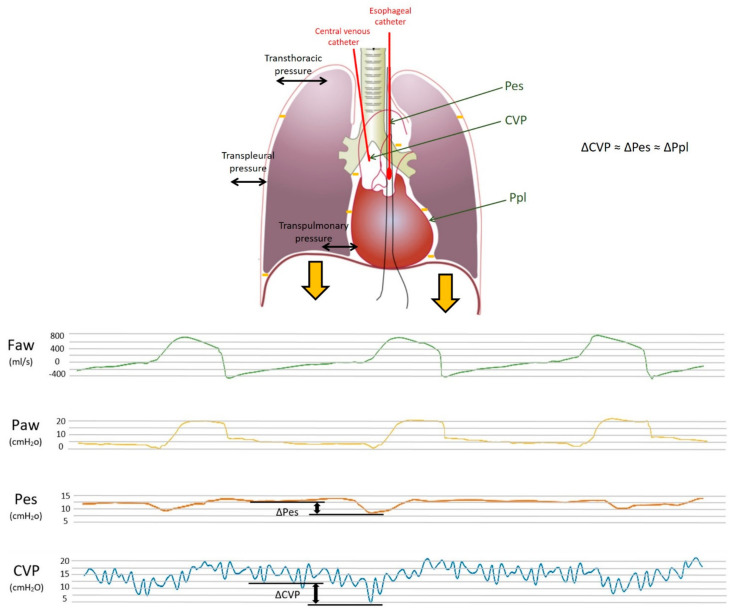
Upper panel: Model of the respiratory system and transmission of pleural pressure during inspiration. The respiratory system is composed of the lungs and the chest wall in series. The figure shows the different pressures within the system and the relative distending forces (in black). The difference between the alveolar and atmospheric pressure, i.e., the transthoracic pressure, is the pressure that distends both the lungs and the chest wall; the transpleural pressure (i.e., pleural minus atmospheric) is the pressure needed to distend the chest wall, whereas the transpulmonary pressure (i.e., airway minus pleural) is the pressure that distends the lungs. The thick, orange arrows depict the downward displacement of the diaphragm during inspiration, which lowers the pleural pressure (orange minus signs). This negative pleural pressure swing (ΔPpl) diffuses into the intrathoracic space and is transmitted through the esophagus to the balloon-tipped esophageal catheter (ΔPes) and through the superior vena cava to the central venous catheter (ΔCVP). Pes, esophageal pressure; Ppl, pleural pressure; CVP, central venous pressure. Lower panel: Pressure waveforms for CVP, Pes and Paw. Central venous pressure swings (CVP, blue wave), esophageal pressure swings (Pes, orange wave), airway pressure swings (Paw, yellow wave) and flow (Faw, green wave) during assisted mechanical ventilation.

**Table 1 diagnostics-13-01022-t001:** Factors influencing the relationship between CVP and pleural (Ppl) or esophageal (Pes) pressure.

Factors Favoring the Use of CVP	Factors Limiting the Use of CVP
Homogeneous transmission of intrathoracic pressure into the mediastinum	Fluid (CVP) vs. air (Pes) filled systems, different lengths and diameters of the catheters
Good correlation of tidal swings of CVP and Pes	Different absolute value between CVP and Pes
Widespread use of central venous catheters	Attenuated transmission of pressures, amplified with increasing values of CVP
High compliance of the superior vena cava	Temporal delay between CVP and Ppl due to nonhomogeneous distribution of lung pressures
High accuracy of modern pressure transducers	Presence of cardiac component and artifacts
No or little effect of baseline CVP values over CVP swings	The CVP/Pes relationship is unpredictably influenced by body position

**Table 2 diagnostics-13-01022-t002:** Main studies regarding the use of CVP swings as an estimate of esophageal or pleural pressure swings.

Author, Year	Study Type	Sample	Ventilation	Findings	Conclusions
Walling and Savege, 1976 [31]	Clinical observational	9 pts	CMV	ΔP_L_ measured from ΔPes was greater than ΔP_L_ measured from ΔCVP. This increase was similar to that of ΔPes compared with ΔPpl previously reported.	If ΔPes shows a similar increase over both ΔPpl and ΔCVP, these should be nearly equal. ΔCVP could be the best and least invasive estimate of ΔPpl during ventilation.
Kyogoku et al., 2020 [32]	Clinical observational	7 pts	CMV	Comparing ΔPes, ΔCVP and ΔPpl calculated using a corrected ΔCVP (cΔCVP-derived ΔPpl), the latter correlated better with ΔPes than did ΔCVP.	This correction method improved the accuracy of ΔPpl estimation by using ΔCVP during controlled ventilation in children.
Okuda et al., 2021 [33]	Clinical observational	8 pts	AMV	Comparing ΔPes, ΔCVP and ΔPpl calculated using a corrected ΔCVP (cΔCVP-derived ΔPpl), the latter correlated better with ΔPes than did ΔCVP.	This correction method improved the accuracy of ΔPpl estimation by using ΔCVP during assisted ventilation in children.
Verscheure et al., 2017 [34]	Clinical observational	30 pts	CMV/AMV	Comparing ΔPes and ΔCVP, the bias was close to 0 mmHg in CMV and −2 mmHg in PSV.	ΔPes tracks both positive and negative changes in Ppl as indicated with ΔCVP.
Flemale et al., 1988 [35]	Clinical observational	10 pts	SB	Comparing ΔCVP, ΔPes and ΔPm at occlusion test (taken to represent ΔPpl), ΔCVP/ΔPm, ΔPes/ΔPm and ΔPes/ΔCVP were close to unity.	ΔCVP and ΔPes could provide, in most instances, accurate measurements of ΔPpl during inspiratory efforts with occluded airways.
Aguilera et al., 2018 [36]	Clinical observational	9 pts	SB	Measuring cough pressure with Pes and comparing it with Pga, CVP, Pbl and Prec, median maximum pressures at those different sites were similar and agreed well with Pes.	Less invasive catheters can measure cough pressure. Intrathoracic pressure changes are well transmitted to the vascular compartment and, in particular, to the SVC.
Chieveley-Williams et al., 2002 [37]	Clinical observational	10 pts	AMV	ΔCVP and ΔPbl correlated with ΔPes and ΔPga. Their changes were comparable when inspiratory pressure support was reduced.	Despite the ΔCVP absolute value does not perfectly reflect ΔPes, ΔCVP reveals an increased patient inspiratory effort during support reduction.
Biselli and Nobrega, 2017 [38]	Clinical observational	10 pts	AMV	ΔCVP had a good performance for measuring work of breathing and intra-thoracic pressure swings when compared to ΔPes.	CVP could be an easy and bedside method for assessing patient inspiratory effort and ventilatory mechanics.
Colombo et al., 2020 [39]	Clinical observational	48 pts	AMV	ΔCVP well identified strong inspiratory efforts and showed a good agreement and correlation with ΔPes at ZEEP and 10 cmH_2_O CPAP.	ΔCVP, even if sometimes differing from ΔPes exact value, helps to identify strong inspiratory efforts and titrate support.
Lassola et al., 2021 [40]	Clinical observational	14 pts	AMV	ΔCVP was significantly associated with ΔPes during a three-level pressure support trial, and ΔCVP could discriminate high inspiratory efforts.	ΔCVP might be a rapid bedside tool to monitor patient inspiratory effort during assisted mechanical ventilation.
Ostrander et al., 1977 [41]	Physiological observational	10 dogs	SB	Comparing ΔCVP and directly measured ΔPpl, ΔCVP was approximately 55% of ΔPpl at low mean CVP and decreased to 20% of ΔPpl at high mean CVP.	Caution is needed when attempting to evaluate respiratory ΔPpl directly from ΔCVP because of the presence of many distortions.
Hedstrand et al., 1976 [42]	Physiological observational	13 sbj	SB	ΔCVP and ΔPes showed a strong mean phase divergence, and the mean quotient ΔCVP/ΔPes varied significantly, even with body position.	ΔCVP could not simply replace ΔPes in estimation of ΔPpl during breathing.
Hylkema et al., 1983 [43]	Clinical observational	12 pts	CMV	ΔPes tended to be greater than ΔCVP, and these measurements showed no correlation during controlled ventilation.	ΔCVP does not exactly replicate ΔPes, which seems to remain the best choice in estimating tidal ΔPpl.
Bellemare et al., 2007 [44]	Clinical observational	24 pts	AMV	Average inspiratory ΔCVP and average inspiratory ΔPes had low correlation during spontaneous respiratory efforts.	ΔCVP is not a good predictor of ΔPpl during spontaneous respiratory efforts.

CMV (controlled mechanical ventilation), SB (spontaneous breathing), AMV (assisted mechanical ventilation).

**Table 3 diagnostics-13-01022-t003:** Biases and limits of agreement of the studies regarding the use of CVP swings as an estimate of esophageal or pleural pressure swings.

Author, Year	Biases	Limits of Agreement
Walling and Savege, 1976 [31]	Small sample size, indirect conclusions, different measurement systems (fluid-filled and air-filled).	Pl swings measured from ΔPes were 30% higher than those measured from ΔCVP. Previous studies reported a similar higher ΔPes (26.4%) when compared with the direct measurement of ΔPpl, suggesting a comparable value of ΔPpl and ΔCVP.
Kyogoku et al., 2020 [32]	Small sample size, included only children, severe ARDS cases not included, most cases post cardiac surgery.	cΔCVP-derived ΔPpl correlated with ΔPes better than did ΔCVP (R^2^ = 0.48, *p* = 0.083 vs. R^2^ = 0.14, *p* = 0.407)
Okuda et al., 2021 [33]	Small sample size, included only children, impact of cardiogenic oscillations on ΔCVP and ΔPes measurements.	Difference of cΔCVP-derived ΔPpl to ΔPes was smaller than that of ΔCVP to ΔPes at all support levels (−0.1 ± 1.5 vs. 3.1 ± 3.5 cmH_2_O in PS 10, −0.7 ± 3.3 vs. 4.5 ± 3.9 cmH_2_O in PS 5, and −1.0 ± 3.4 vs. 4.7 ± 4.4 cmH_2_O in PS 0)
Verscheure et al., 2017 [34]	Effect of gravity and frequency response of fluid-filled catheters, ventilator-triggered breaths during controlled ventilation.	Comparing ΔPes and ΔCVP, the bias was close to 0 mmHg in CMV and −2 mmHg in PSV.
Flemale et al., 1988 [35]	Small sample size, cardiac artifacts with fluid-filled esophageal catheters, hydrostatic pressure gradient between catheter tip and pressure transducer modified by respiratory movements.	ΔCVP/ΔPm, ΔPes/ΔPm and ΔPes/ΔCVP individual values and group average were close to unity in all positions.
Aguilera et al., 2018 [36]	Small sample size, measurements limited to cough, no ventilation.	Average maximum pressures at different sites (gastric, bladder and rectal) were similar, and an excellent agreement was found between alternative sites and Pes.
Chieveley-Williams et al., 2002 [37]	Small sample size, different measurement systems (fluid-filled and air-filled), relative and absolute positions of the catheters, different pathologies, varying frequency response of the systems with time.	ΔPes/ΔCVP ratio varied between 0.8 and 2.1, and the ΔPga/ΔPbl ratio varied between 0.6 and 1.3. Reducing ventilator assistance, the variation in ΔCVP was related to the variation of ΔPes.
Biselli and Nobrega, 2017 [38]	Small sample size, cardiac artifacts, high variability of inspiratory compliance.	ΔCVP highly correlated with ΔPaw during the Muller maneuver, comparable to that between ΔPes and ΔPaw. CVP had good performance for measuring WOB (R^2^ = 0.89) and intrathoracic pressure swings (R^2^ = 0.75) compared to Pes.
Colombo et al., 2020 [39]	Stable hemodynamics and no clear evidence of hypervolemia, early and severe ARDS were not included, no comparison between ΔCVP and diaphragm electrical activity or thickening.	ΔCVP identified strong inspiratory efforts with an area under the curve >0.9 both at ZEEP and during CPAP.
Lassola et al., 2021 [40]	Small sample size, limited timeframe, pressure support at enrolment set by the clinician, transdiaphragmatic pressure and intrinsic PEEP not measured, only COVID-19-related acute respiratory failure.	Reducing support, ΔPes and ΔCVP increased in a similar way (5 [3; 8] vs. 8 [14; 13] vs. 12 [6; 16] and 4 [3; 7] vs. 8 [5; 9] vs. 10 [7; 11] cmH_2_O, respectively); ΔCVP was significantly associated with ΔPes with R^2^ = 0.810.
Ostrander et al., 1977 [41]	Small sample size, non-clinical physiological study.	ΔCVP was 55% of ΔPpl at low mean CVP and 20% of ΔPpl at high CVP values.
Hedstrand et al., 1976 [42]	Small sample size, different measurement systems (fluid and air-filled), physiological study.	ΔCVP/ΔPes ratio varied significantly: 0.28 (0.12–0.44) in supine, 0.42 (0.25–0.56) in semirecumbent and 0.68 (0.34–1.40) in seated position.
Hylkema et al., 1983 [43]	Small sample size, different measurement systems (fluid and air-filled).	ΔPes was higher than ΔCVP, and the measurements only weakly correlated (r = 0.47).
Bellemare et al., 2007 [44]	Different right and left heart filling volumes could influence ΔCVP and ΔPes correlation.	Bias between ΔCVP and ΔPes was 2.9 cmH_2_O with R^2^ = 0.43.

## Data Availability

Not applicable.

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
