# Peer review of "Respiratory Variations of Central Venous Pressure as Indices of Pleural Pressure Swings: A Narrative Review"

_diagnostics, 2023, doi:10.3390/diagnostics13061022_

Round 1

Reviewer 1 Report

The authors have made a review about the usage of CVP for pleural pressure and they compared it with the oesophageal gold standard measurements. I think this could simplify the intrapleural pressure measurement. Please advise how it can be used in daily practice with the available monitors. It would be also interesting to summarize the mean bias and limit of agreement in one table.

Two minor comments: please use PAOP or PAWP for the PAC measured wedge pressure 2, chest compression is usually used for cardiopulmonary resuscitation please rephrase. 

Author Response

The authors have made a review about the usage of CVP for pleural pressure and they compared it with the oesophageal gold standard measurements. I think this could simplify the intrapleural pressure measurement. Please advise how it can be used in daily practice with the available monitors. It would be also interesting to summarize the mean bias and limit of agreement in one table.

R: thanks for the interesting comments. A new paragraph has been added to explain how to use the CVP swing in the daily practice. It reads: “On a practical standpoint, we suggest analyzing the CVP trace using the “wedge pressure procedure” available in almost any standard intensive care multiparametric monitor. This procedure, in fact, displays an invasive pressure trace together with the respiration waveform as a reference, and allows the clinician to freeze the trace and move the cursor while displaying the actual pressure value, allowing for an accurate selection of the inspiratory and expiratory values of CVP relative to the respiratory cycle.”

A new table (Table 3) has also been added with mean bias and limits of agreements of the different studies

Two minor comments: please use PAOP or PAWP for the PAC measured wedge pressure 2, chest compression is usually used for cardiopulmonary resuscitation please rephrase. 

R: The text has been amended as suggested

Reviewer 2 Report

The article titled "Respiratory variations of central venous pressure as indices of pleural pressure swings" presented by the authors is quite interesting and I think it is important for readers. The article may be accepted for publication.

Author Response

The article titled "Respiratory variations of central venous pressure as indices of pleural pressure swings" presented by the authors is quite interesting and I think it is important for readers. The article may be accepted for publication.

R: thanks for the interest and the comments

Reviewer 3 Report

The authors submitted an interesting review on respiratory variations of CVP as surrogate indices of pleural pressure variations. I see no major flaws in the manuscript, but points of improvement may be a discussion of limitations of the CVP and the addition of a figure showing lungs, pleural space, and central veins and providing a schematic summary of what is explained.

In particular, I see no mention of how fluid balance and cardiac failure (e.g. especially right ventricle) can influence CVP sensitivity and variations. Regarding the figure, it is of course not mandatory, but it can increase the visibility of the article.

Author Response

The authors submitted an interesting review on respiratory variations of CVP as surrogate indices of pleural pressure variations. I see no major flaws in the manuscript, but points of improvement may be a discussion of limitations of the CVP and the addition of a figure showing lungs, pleural space, and central veins and providing a schematic summary of what is explained.

In particular, I see no mention of how fluid balance and cardiac failure (e.g. especially right ventricle) can influence CVP sensitivity and variations. Regarding the figure, it is of course not mandatory, but it can increase the visibility of the article.

R: the reviewers underlined an interesting and missing issue and suggested a nice modification of the figure. A new paragraph has been added in the limitations section, expanding what was already written about fluid status and including a new part on right heart function, with new references.

Figure 2 was modified introducing a new upper panel showing a scheme of the thorax with lungs, heart and central veins, the esophagus and the relative pressures during spontaneous inspiration.